# Enhancing Episiotomy Skills Through Interactive Online Simulation

**DOI:** 10.3390/healthcare13121472

**Published:** 2025-06-18

**Authors:** Hülya Tosun, Hava Özkan

**Affiliations:** 1Midwifery Department, Health Science Faculty, Kutahya Health Science University, 43700 Kütahya, Türkiye; 2Midwifery Department, Health Science Faculty, Atatürk University, 25240 Erzurum, Türkiye; havaoran@atauni.edu.tr

**Keywords:** online interactive simulation, episiotomy experiences, midwifery, pandemic

## Abstract

**Background/Objectives:** The COVID-19 pandemic has significantly restricted clinical training for midwifery students, highlighting the need for alternative teaching methods. With the disruption of traditional face-to-face education, online simulation-based training has emerged as an effective alternative for developing essential clinical skills. The acquisition of hands-on skills has a direct impact on students’ self-confidence and clinical performance. Interactive online simulations support the learning process by enhancing both theoretical knowledge and practical competencies. This study aims to evaluate the impact of online simple simulation-based episiotomy repair training on students whose clinical practice was limited due to the COVID-19 pandemic. **Methods:** A mixed-method approach was used, considering the outcomes from 61 midwifery students. Data were collected via observational questionnaires, which provide an online learning readiness scale and scales for student satisfaction and self-confidence. The analysis included descriptive statistics, McNemar’s, binary logistic regression, and the Mann–Whitney U test. **Results:** Students who trusted themselves in both opening and repairing an episiotomy after training had more readiness for online learning (t(43) = 2.73, *p* = 0.009; t(43) = 2.40, *p* = 0.02). Students with better training performance are more likely to obtain higher scores on the final exam of the Clinical Practice module (rho = 0.33, *p* = 0.01). Additionally, their performance was a positive and significant predictor of achieving a full mark (b = 0.11, s.e. = 0.05, *p* = 0.01). **Conclusions:** Interactive online simulation training improved midwifery students’ hand skills and self-confidence in clinical practice. Such methods should be promoted in circumstances like COVID-19.

## 1. Introduction

Simulation-based practical training has become fundamental to modern midwifery training, providing dynamic possibilities to connect theoretical instruction with clinical practice [1]. The World Health Organization (WHO) has made a global and strategic recommendation to nursing and midwifery educators [2]. This recommendation emphasizes that educators should train graduates capable of addressing the specific health needs of their communities.

Simulation models, whether classical or technologically modern, offer structured environments for learners to safely cultivate essential clinical competencies [3]. These models facilitate the integration of cognitive, psychomotor, and affective domains while simultaneously augmenting students’ self-confidence, critical thinking, and problem-solving skills through experiential engagement [1,3,4]. Clark [5] asserted that these instructional methods strongly correspond with constructivist learning theory, wherein knowledge is actively generated through interaction and reflection. In this setting, simulation training—spanning from low-fidelity anatomical models to high-fidelity immersive systems—serves a crucial function in providing equitable and engaging clinical learning experiences. Alinier et al. [6] classified simulation into six categories, ranging from written simulations (Level 0) to interactive patient simulators (Level 5), offering a thorough framework for comprehending the educational significance of different simulation types. Although modern models provide multisensory, high-precision learning, resource-constrained alternatives, such as animal tissue-based practice, retain significance due to their tactile feedback and anatomical realism [7,8,9].

The COVID-19 pandemic marked a significant turning point for global healthcare education, revealing the vulnerability of a curriculum reliant on in-person instruction [10]. However, the sudden shift to online education led to a marked reduction in practical clinical experience, affecting midwifery students’ readiness for professional practice [11,12,13].

While universities’ digital platforms provided temporary solutions for theoretical continuity, the deficiencies in obtaining practical skills via virtual interfaces heightened apprehensions among senior students regarding their preparedness for the workforce [14,15]. This disruption initiated a pedagogical change, prompting institutions to investigate creative methods to maintain clinical learning outcomes.

Utilizing Kolb’s experiential learning theory, which highlights concrete experience and reflective observation [16], alongside Bandura’s social cognitive theory that emphasizes the significance of self-efficacy in learning [17], educators have increasingly adopted interactive simulation to maintain educational quality during the crisis. These methods provide experiential depth and correspond with comprehensive crisis management frameworks in education, fostering adaptive learning pathways amid systemic shocks [18]. Simulation has evolved into a strategic pedagogical instrument, transcending mere substitution for clinical practice, to guarantee the continuity, accessibility, and integrity of midwifery education during crisis [19].

This study expands on the existing theoretical and practical framework by investigating the impact of interactive online simulation on midwifery students’ development of episiotomy healing abilities. In particular, its objectives can be encapsulated under the headings of assessing the alterations in students’ perspectives and attitudes towards the efficacy and necessity of episiotomy practice along with their self-assurance in their competencies before and during the training, attaining a comprehensive understanding of how individual variances among students enhance the efficacy and satisfaction of online episiotomy techniques and experiences, and investigating the extent to which the training enhances academic achievement.

## 2. Materials and Methods

This study is semi-experimental, and it has an observational design. It was performed among 4th-grade students (n = 80) from the Midwifery Department at Kutahya Health Science University -Erzurum. The ethics committee for the study was obtained from The Ethics Committee of Faculty of Health Sciences, Atatürk University, with the date of 12 February 2021 and number of 2022/02/03. The research was conducted in 2021, between 20 February and 10 March.

In alignment with these objectives, the following research questions in terms of the benefits of training are posed under the predefined subgroups: the differences across all parameters of the pre- and post-training questionnaires and the manual skills improved through online education, considering the period before and after the training (Table 1).

An interactive online simulation study group within the scope of the Clinical Practice course for midwifery senior students at Kutahya Health Science University was conducted, while the “purposeful sampling” (non-probability sampling) method was chosen due to time and location constraints.

### 2.1. Participants

Final-year students enrolled in a university midwifery program who voluntarily agreed to participate, had access to the standard equipment and conditions required for the practice sessions, and completed all phases of the study without interruption were included in the research. Participation was entirely voluntary, and students were informed of their right to withdraw from the study at any time without penalty.

Although a total of 80 students initially participated in the study, the analysis was conducted with data from 61 students who were able to perform the procedure on live tissue. The rationale for focusing on live tissue was to provide a more realistic and hands-on experience in episiotomy repair.

Nineteen students were excluded from the analysis, as they did not meet the material-related eligibility criteria. These students, facing financial constraints, were unable to obtain live tissue materials and instead conducted the procedures using alternative resources such as sponges or synthetic leather. Nevertheless, their participation in the practical sessions was encouraged in accordance with the principle of equity in education.

It is noteworthy that approximately 80% of the participants belonged to middle or lower socioeconomic backgrounds, which may limit their access to appropriate training materials. This financial disparity highlights a significant challenge in ensuring uniformity in educational opportunities and may necessitate the exclusion of some students from data analysis to maintain methodological rigor. In line with this rationale, students who were unable to access the required materials were excluded from the final analysis.

The university’s midwifery department is only able to provide laboratory materials for in-person practical sessions and is not in a position to offer additional financial support or distribute materials to students individually.

Students were contacted via e-mail through the University Student Information System one month before the training. They were lectured on how to perform an episiotomy. They were informed about the content of the study, and a pre-training survey (pretest) was administered one week before the training (Appendix A). Episiotomy videos and episiotomy instructions developed by Illston et al. [9] were sent to the students. The group simulation study was formed, and the participants were informed that a passing grade was not guaranteed. They were prepared in virtual classrooms with pre-planned practice guides with the guidance of an academician to increase the knowledge and skills of making incisions and suturing in living tissues, such as calf tongue.

#### Power Analysis

A priori power analysis was conducted based on a one-sample *t*-test design, using Cohen [20] conventions. Assuming a medium effect size (Cohen’s d = 0.5) with an alpha level of 0.05 and a total sample size of 61 participants, the calculated statistical power was 0.97. The result indicates that the study has a 97% probability of detecting a medium effect, suggesting the sample size is sufficient for reliable results.

### 2.2. Procedure

Due to limited resources, the simulation methodologies in this study focused on levels 1 and 2 of Alinier’s et al. [6]. In addition, it is cost-effective and can be performed with a certain number of students and an academic person. The equipment is also portable and available, so students can repeat it as well as feedback on performance is easy to obtain [6].

The training was carried out in an four online interactive training groups (three groups of 15 people and one group of 16 people). The four groups had interactive practice in separate periods, while the duration of each training was 60 min, for a total of 240 min. This training was recorded to check whether the evaluations were objective or not as well as to provide feedback to the students.

Two external experts, who were from different institutions, observed each student’s practice. One of them was from a different university midwifery department, and the other one was from a state hospital delivery room. The experts guided the students about their actions during the training. At the end of these practices, a posttest was applied to the students for evaluation (Table 2).

#### 2.2.1. Creating an Interactive Online Education Environment

The trainer installed a camera at home and demonstrated how the visual feed would be streamed to students in real time via the Internet. These sessions were recorded through the Distance Education Module of the University Information Management System, which serves as an existing course delivery platform. This system was widely utilized by the university, particularly during the COVID-19 pandemic. The camera was 40 cm above the study area. The environment was bright, and the interference could be observed clearly with details. When it was ensured that there was no audio the training started.

#### 2.2.2. Preparation of Training Materials

For the training, half-frozen animal tissue was used. As the episiotomy set, a blunt, curved type surgical scissors (12.5 cm), pivot (15 cm), forceps (14 cm), scalpel handle, blade, 3-0 or 2-0 absorbable suture, syringe, gloves, mask, and sponge were used. The moulage blood technique was used to create a more realistic and natural clinical environment (red beet as well as tomato paste or ketchup was recommended).

#### 2.2.3. Episiotomy Simulation Instructional Process

The educational session began with the assembly of all necessary supplies, including the episiotomy set and supplementary instruments. Animal tissue, previously cryopreserved, was permitted to thaw at ambient temperature until it attained a manageable consistency. Upon preparation, the tissue was firmly placed on the work surface.

Participants engaged in role play to administer local anesthetic, simulating the clinical scenario. The episiotomy site was subsequently washed and disinfected with a sterile solution. Using a scalpel, a vaginal opening measuring roughly 4 to 5 cm was produced to simulate a more realistic clinical situation. The technique was conducted on the muscular segment of the tissue model.

Later, a mediolateral incision of three centimeters was created on the right side, around two centimeters above the vaginal terminus. The depth of the incision was altered so that it simulated both first-degree and second-degree incisions. An accurate staining material was used to generate a simulation of bleeding, which was performed in order to make the experience more realistic (Figure 1). After receiving this explanation and watching the demonstration, the participants independently performed the steps in order to practice the proper technique.

#### 2.2.4. Suture Repair Protocol

During the phase of suture repair, a step-by-step method was utilized, and the individual was provided with guided training [21]. The participants chose an acceptable suture material, which was typically an absorbable suture having a size of 3-0 or 2-0, which is ideal for the repair of soft tissue.

Approximation and suturing of the bulbocavernosus muscle, transverse perineal muscles, and the perineal membrane were the first steps in the process of repairing the damage. If the vaginal wall had a tear, it was sutured at the hymenal level to restore the continuity of the anatomical structure.

Following that, a subcuticular approach with intradermal sutures was utilized in order to repair the perineal skin. This was carried out in order to guarantee the best possible healing without leaving any scars. In conclusion, there was a simulated examination of the wound site carried out by the participants. This assessment included verifying the integrity of all sutures and looking for potential bleeding. This assessment was carried out through role play.

### 2.3. Collecting Data

The data from the following forms were obtained from the students.

#### 2.3.1. Personal Information Form

The information about participants’ age, living place, access to the Internet, and previous experience with living tissue was obtained.

#### 2.3.2. Pre- and Post-Education Questionnaire

The researchers prepared a form by considering the studies in the literature [4,9,22,23,24]. The pre-education form contained seven questions, and the post-education form included seven (Appendix A).

#### 2.3.3. Online Learning Readiness Scale (OLRS)

This is a self-report type measurement tool used to determine students’ readiness for online learning [25]. It was adapted by İlhan and Çetin [26] with a 5-point Likert-type scale consisting of 18 items as well as a 5-factor structure. In their study, the Cronbach’s alpha of the scale was 0.87, while it was the same in this study.

#### 2.3.4. Student Satisfaction and Self-Confidence in Learning Scale (SCLS)

This is used to measure students’ attitudes and beliefs about simulation [27]. It was adapted by Karaçay and Kullanılan [28] with a 5-point Likert type consisting of two sub-dimensions called satisfaction with learning and self-confidence, consisting of 13 items. The highest score, obtained from the sum of the items, from the scale is 65, while the lowest score is 13. The Cronbach’s alpha of the scale was 0.94, whereas it was 0.90 in this study.

### 2.4. Episiotomy Simulation Skills Rubric

The performance of the students was evaluated using a rubric where each criterion is graded on and pointed from 1 and 4, where the lowest score is 1, and the highest is 4. (Table 3).

### 2.5. Evaluation of Data Obtained Through Observation

Both the instructor and observers were present during the training sessions. The students’ skills and achievements throughout the online episiotomy training were assessed considering the data collected. Each student received a performance score ranging from 0 to 4 based on their execution in the episiotomy simulation (Table 3).

The objectivity and reliability of the observers’ evaluations were assessed using the interclass correlation coefficient (ICC). To measure the consistency of the observers’ evaluations of students’ skills during the episiotomy training, a two-way mixed-effects model with average measures and consistency type was applied. The resulting ICC value was 0.89, with a 95% confidence interval (CI) of [0.82, 0.94], indicating a high level of agreement between the observers. The results suggest that the students’ performance during the episiotomy simulation was evaluated consistently among the raters [4].

The average grades obtained by students in this course (Clinical Practice course) were included in further analyses to examine whether there was a relationship between students’ academic performance and their clinical practice competencies.

### 2.6. Statistical Method

Analyses were performed by using IBM SPSS Statistics 23 with the G*Power package program (Version 3.1) for the power analysis [20]. The results indicated that a sample size of 61 students was sufficient for the power analysis. A significance level of 0.05, a power of 0.85, and an effect size of 0.91 were applied in the analysis. Regarding the evaluation of parameters in the pre- and post-education forms, frequencies (number and percentage) for categorical variables and descriptive statistics (mean and standard deviation) for numerical variables are provided. McNemar’s analysis was used to compare the changes in two dependent categorical variables between pre- and post-training questionnaires. The data were not normally distributed, so non-parametric correlations between variables were examined. Mann–Whitney U tests and an independent *t*-test were performed while investigating the group differences in students’ self-confidence, satisfaction with learning, and online readiness between students who were eager to open and repair an episiotomy. Finally, binary logistic regression was performed between the observational data and a later categorical version of the final mark in the module [29].

## 3. Results

The content of Table 4 was obtained from personal information. It shows that most of the participants (50.8%) live in the city, and more than 50% of them had good or very good Internet access. The rate of those who had sutured living tissue before was 23.0%, while it was 4.9% for those who undertook episiotomy repair in the delivery room. In addition, 67.2% of them were interested in clinical practice simulations.

Table 5 presents the descriptive statistics between the SCLS and OLRS. It is worth noting that the average and standard deviation of the OLRS, SCLS, student satisfaction with learning, and self-confidence in learning were 74.96 ± 8.12, 47 ± 10.8, 20.24 ± 4.67, and 26.86 ± 6.46, respectively.

Table 6 presents the relationships between SCLS and OLRS, where student satisfaction and online learning readiness are weakly positive (r = 0.204) and statistically insignificant (*p* = 0.159). On the other hand, the confidence in learning and online learning readiness relationship is moderately positive (r = 0.326) and statistically significant (*p* = 0.022).

### 3.1. The Differences in All Parameters of the Pre- and Post-Training Questionnaire

The students’ answers to the questions before and after training are provided in Table 7. A total of 67.2% before and 90.2% after the training thought that an episiotomy study with simulation performed on living animal tissue could be considered a real episiotomy study. The ratio of those who agreed about this issue increased significantly after the training compared to the pre-training (*p* = 0.003).

A total of 49.2% and 96.7% of the participants, respectively, thought that hand skills gained in online education would be effective thereafter. The result indicates that there was a significant increase after the education (*p* = 0.000). The answers to the question about finding oneself inadequate in suturing significantly decreased from 45.9% to 26.2% after the training (*p* = 0.004), while the answers for the rest of the questions were not statistically significant (*p* > 0.05).

The results of the education evaluation are presented in Appendix B. About 96.7% of the participants agreed on the benefit of the interactive online simulation training, while 60.6% of them believed that interactive online simulation training increased the objectivity of the training. The participants thought that the online training was useful in improving their manual skills (96.7%). Moreover, 100% of them expressed interest in attending such interactive online simulation training in the case that they were obliged to attend online training, such as in a disaster or crisis.

### 3.2. Determination of the Students’ Benefit from the Training

The parameters of observed performance during training, online readiness scale, student learning satisfaction, student self-confidence, and marks in Clinical Practice after training were cross-evaluated (Table 8). Since the skewness and kurtosis values exceeded ±1.5 [30], the variables of student self-confidence and student learning satisfaction were not normally distributed, so the Spearman rho correlation was preferred to evaluate them.

A moderate positive correlation was determined between observed performance during training and the Clinical Practice module grade after training (rho = 0.327; *p* = 0.011), indicating that better performance during training is associated with higher Clinical Practice module grades afterward.

There is a moderate positive correlation between online readiness and student learning satisfaction (rho = 0.393; *p* = 0.008). Online readiness was also found to be moderately correlated with students’ self-confidence (rho = 0.496; *p* = 0.001), suggesting that students who feel more prepared for online learning are more confident in their abilities. In addition, the correlation between online readiness and Clinical Practice module grades is weak and not statistically significant.

A strong positive correlation was obtained between student learning satisfaction and student self-confidence (rho = 0.844; *p* < 0.001), indicating that students who are more satisfied with their learning experiences are more confident.

The correlation between student learning satisfaction and Clinical Practice module grades is weak and not statistically significant. In addition, there is also no significant correlation between student self-confidence and the Clinical Practice module grade after training (rho = 0.052; *p* = 0.768).

To investigate whether there are any significant group differences between students who trusted themselves to open and repair an episiotomy after training and those who did not, a Mann–Whitney U test for not-normally distributed variables (self-confidence and satisfaction in learning) and an independent *t*-test for the normally distributed score of online readiness were applied. No significant group differences were obtained in student confidence and satisfaction between the two groups (U = 248, *p* = 0.935; U = 222, *p* = 0.500; U = 221, *p* = 0.556; U = 236, *p* = 0.796). It was significantly determined that students who trusted themselves in both opening and repairing an episiotomy after training had more readiness for online learning (t(43) = 2.73, *p* = 0.009; t(43) = 2.40, *p* = 0.02).

### 3.3. Relationship Between the Performance During the Episiotomy Simulation and Marks

Regression analysis was performed to check the final exam marks and the students’ performance during episiotomy training. Skewness and kurtosis values indicated that both variables are normally distributed, but the homogeneity of variance was not met since final marks were varied on a limited range, where 39 students achieved full marks (100) while 21 of them had between 80 and 96. Therefore, a categorical variable was created to represent a “full mark” (coded as 1) or “not full mark” (coded as 0) to perform a binary logistic regression to ascertain the relationship between the two variables. The results showed that the model was statistically significant concerning the parameters of χ^2^(1) = 10.88 and *p* = 0.001. It explained 24.0% (Nagelkerke R^2^) of the variance in achieving full marks in the final exam and correctly classified 71.7% of cases [31].

Observed performance was also a positive and significant (b = 0.11, s.e. = 0.05, *p* = 0.01) predictor of the probability of earning a full mark. The results indicated that an increase in observed performance (odds ratio =1.115, 95% CI [1.02, 1.22]) during episiotomy simulation significantly increased the odds of obtaining a full mark.

## 4. Discussion

The COVID-19 pandemic significantly affected the education system by forcing institutions to use online learning [32]. The results of this study reveal that students perceived online education as a successful, effective, and easily accessible process that builds skills and self-confidence for midwifery. Specifically, students’ OLRS total scores and their self-confidence in learning sub-dimension scores were high and found to be correlated. Compared to another study in which nursing students’ expectations and readiness for online learning during the pandemic were investigated, the obtained students’ OLRS scores were concluded to be high [33]. These findings align with observations from other studies that reported high levels of student satisfaction with online learning during the pandemic [34].

These positive outcomes may be due to increased interaction with the computer technologies of this era [35]. Additionally, the students had good Internet access and a moderate economic situation level, which is considered an important factor in enabling online learning.

In addition, higher SCLS scores indicated that advanced life-support interventions were effectively performed on a simulated patient presenting with myocardial infarction using a high-fidelity simulation, as shown in the study by Karahan et al. [36]. The students reported increased self-confidence and expressed satisfaction with the training carried out using a high-reality patient simulator.

Similarly, Kiraz et al. [37] implemented skills training with a highly valid simulator and found that students’ anxiety levels significantly decreased in the anxiety sub-dimension of the learning attitudes scale, which could positively influence learning attitudes.

Demirel et al. [24] conducted a face-to-face bovine tongue episiotomy repair simulation training study, revealing decreased anxiety levels and increased self-efficacy among students. These outcomes consistently highlight that simulation training contributes to reduced anxiety, enhanced self-confidence, and improved learning by providing accessible, repeatable, and interactive educational environments.

Furthermore, 96.7% of the students in the present study found the interactive online simulation training method to be effective and beneficial (Table 7). Supporting this, Prasad et al. [15] conducted a perinatal emergency intervention simulation study among medical and midwifery graduates and reported that students found the e-learning format meaningful and valuable for interprofessional skills development.

In summary, multiple studies emphasize that both face-to-face and online simulation training enhance students’ learning experiences by lowering anxiety, boosting self-confidence, and promoting active engagement through accessible, repetitive, and communicative environments.

The before- and after-training comparison indicated statistically significant results. The proportion of those who said that “an experiment should be performed on animal tissue before a real episiotomy study” at the end of the training increased significantly. In a similar study comparing the effectiveness of surgical interventions on living tissue and simulators for war injuries, it was revealed that experimenting with living tissue models was more appropriate than the original approach [38]. For this reason, healthcare professionals’ preference for trauma training with living tissue in simulators showed that living tissue studies should be performed more widely since they allow students to have a more realistic experience [39].

The results in this study showed that students’ beliefs about their hand skills and suturing proficiency improved following the training. A similar study showed that veterinary students who performed a spaying simulation on live tissue experienced a tenfold increase in self-confidence after the surgery. Their anatomical knowledge and skills also improved by 30% [40]. These findings suggest that simulations involving live animal tissue offer an excellent and effective opportunity to boost students’ self-confidence. However, it is important to note that this study was not conducted online but still serves as a significant example of the skills acquired through simulations with live tissues. Another online training session demonstrated that students successfully acquired suturing skills, but they did not think that these skills were explored. There is a lack of direct comparisons in the literature about the results of this study [41]. Therefore, future studies that incorporate self-assessment and feedback from students will be valuable in better understanding their expectations from courses involving online interventional procedures.

The evaluation of such training is considered beneficial for both instructors and participants, particularly during crises, as it motivates them to engage in similar training sessions. Several studies emphasize the positive impact of an independent researcher’s presence on students. For example, a similar study by Rogers et al. [42] highlighted that the role of an observer in online education can serve as an important control factor, contributing to student satisfaction.

The results in this study indicate that a significant moderate positive correlation was determined between the OLRS and SCLS subscales, which include student learning satisfaction and self-confidence in learning. It suggests that readiness for online learning enhances students’ self-confidence and satisfaction with their learning experience. In a similar study, Oducado [33] reported a significant positive correlation between students’ online learning outcomes and their readiness for online learning. Additionally, Wei and Chou [43] found that online readiness positively influenced students’ learning perceptions and course satisfaction.

These results indicate that when students have access to adequate Internet resources and possess the necessary skills for online education, they approach online training with greater confidence, as their readiness contributes to their satisfaction. Also, Brereton and colleagues explained in their study that practicing using a perineal suture kit at home improved midwifery students’ manual skills and self-confidence [44]. Such outcomes suggest that young people can become more visionary when provided with more technological opportunities.

Students who performed well during episiotomy training were also more likely to achieve higher scores on the final exam. Yukselturk and Bulut [45] reinforced this finding by a significant positive correlation.

Students’ success was also based on their engagement throughout the course. Students who performed well during episiotomy training were more likely to achieve higher scores on the final exam. Yukselturk and Bulut [45] reinforced this finding by a significant positive correlation.

In this study, students’ success was based on their continuous engagement in the course. These students demonstrated a strong sense of responsibility, maintained discipline in completing their tasks, and regularly attended lessons. Even though it was not expected from these students to achieve high final exam grades, it was noted that their motivation increased as they succeeded. Moreover, in e-learning, not only technical skills but also student motivation, online communication skills, and the ability to manage their learning process are very important for academic success [46].

### Limitations

First of all, the small sample size of the study may limit its generalizability. Working with a larger sample size could overcome such problems and contribute to obtaining quite varying results. Additionally, the use of self-reported measures like the OLRS and SCLS could introduce some sort of bias since participants might give socially desirable responses instead of their true opinions. To address the potential bias introduced by self-reported measures like the OLRS and SCLS, future studies could incorporate additional objective assessments, such as performance-based evaluations or peer assessments, to complement self-reported data.

Furthermore, despite the efforts to maintain objectivity, observer bias could have affected student performance, possibly due to the Hawthorne effect. Researchers could use blinded observers who are unaware of the study’s purpose or employ video recordings to allow for a more objective analysis of student performance and minimize the impact of observer bias and the Hawthorne effect.

## 5. Conclusions

The pandemic has highlighted the importance of flexibility and adaptability in educational systems, so this study indicates that online simulations can sustain educational continuity during such crises. To improve student performance, educators should integrate interactive online simulations into the curriculum, especially when clinical training is limited. For instance, providing guided practice, timely feedback, and personalized support for students with low self-confidence can enhance both practical skills and clinical success.

The research presents compelling evidence for the efficacy of interactive online simulation training, especially in crisis periods. By regularly scheduling such training during crises, educational institutions can ensure that students continue to develop their professional skills, maintain engagement, and remain prepared for real-world scenarios.

As we navigate through and beyond the COVID-19 pandemic, these findings offer valuable insights for educators and policymakers. The integration of frequent, high-quality simulation training with traditional clinical experiences presents a robust strategy for maintaining educational excellence, even in the face of unprecedented challenges.

## Figures and Tables

**Figure 1 healthcare-13-01472-f001:**
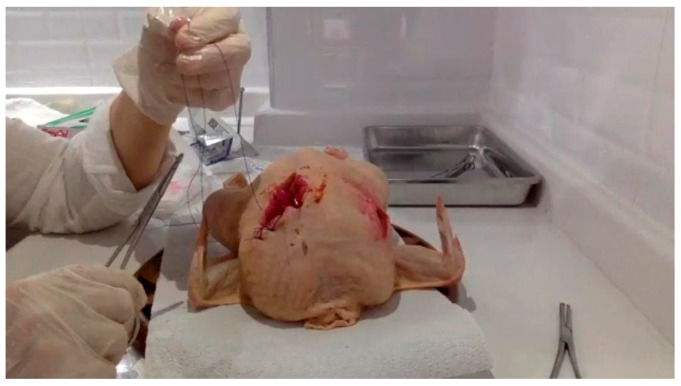
An episiotomy incision and repair practice was carried out in a simulation area arranged by the researcher within their kitchen environment (https://www.instagram.com/p/CKgUfIBFGu9/?igsh=MXdhYXowZGdxNXY0dQ==) (accessed on 11 June 2025).

**Table 1 healthcare-13-01472-t001:** The questions were directed to the online training participants.

Question	Assessment/Outcome Points
(a) Which students benefited the most from the online episiotomy training?	Beneficiary Identification
(b) Do students who are more confident and satisfied with their learning perform better?	Performance and Confidence Relationship
(c) Are students with online readiness more successful?	Online Learning Readiness
(d) Do students who feel more capable of performing and repairing an episiotomy differ in self-confidence, learning satisfaction, and online readiness?	Skill–Confidence Connection
The questions about the outcomes of the training:	
Key Question	Assessment/Outcome Points
(a) Do students who performed better during the episiotomy simulation achieve higher scores in the module afterward?	Effect of Simulation on Academic Achievement
(b) Did the training enhance overall learning in clinical practice and contribute to academic achievement in class?	Overall Learning Impact

**Table 2 healthcare-13-01472-t002:** Simulation training procedure—Flow and timeline.

Stage	Description	Details/ Duration	Responsible Parties
1. Group Allocation	Students were divided into four online groups.	3 groups of 15, 1 group of 16	Two academic instructors
2. Pre-Test Evaluation	Students’ knowledge and skills pre-test.	Before the session, 20 min	Two academic instructors
2. Simulation Session	An interactive online simulation was conducted using basic materials (half-frozen animal tissue).	60 min per group	Two academic instructors, Two external experts (university and hospital)
3. Observation and Rubric Filling	Each student’s performance was observed in real time.	During the session	Two external experts and Two academic instructors
4. Video Recording	Sessions were recorded for objectivity and feedback.	Continuous during practice	University Knowledge Management System (Sisteme Giriş - Kütahya Sağlık Bilimleri Üniversitesi)
5. Feedback and Guidance	Students received formative feedback during practice.	Real time	Two external experts
6. Evaluating	Students practices evaluated	After training	Two academic instructors
7. Posttest Evaluation	Students completed a knowledge and skills posttest.	After the session, 20 min	Two academic instructors

**Table 3 healthcare-13-01472-t003:** The rubric used to evaluate the students in online training.

Criteria	Exemplary (4 Points)	Proficient (3 Points)	Basic (2 Points)	Needs Improvement (1 Point)
1. Preparation of Materials	All materials, including the episode set and auxiliary items, are prepared efficiently and organized systematically.	Materials are prepared adequately with some organization but could be improved.	Materials are prepared but lack organization, leading to inefficiencies.	Materials are incomplete or poorly organized, causing delays or confusion.
2. Tissue Fixation	The tissue is securely fixed to the surface (e.g., tire, clamp, etc.) with no movement during the procedure.	The tissue is fixed securely with minimal movement during the procedure.	The tissue is fixed but has some movement that slightly impacts the procedure.	The tissue is poorly fixed, leading to significant movement and hindering the procedure.
3. Creation of Vaginal Opening	A precise and deep 4–5 cm vaginal opening is created, showing excellent scalpel control.	A 4–5 cm vaginal opening is created with good depth and control.	A vaginal opening is created but lacks depth or precision.	The vaginal opening is shallow or poorly executed, lacking proper scalpel control.
4. Mediolateral Incision	A deep and precise 3 cm mediolateral incision is made 2 cm above where the vagina ends.	A 3 cm mediolateral incision is made with good depth but could be more precise.	The incision is made but lacks depth or precision.	The incision is shallow or not accurately placed, lacking proper depth.
5. Wound Staining	The wound is effectively stained to simulate blood using the appropriate material.	The wound is stained, but the effect could be more realistic.	The wound is stained, but the application is uneven or unrealistic.	The wound is poorly stained or not stained at all, lacking realism.
6. Explanation of Anatomical Structures	Clearly explains and identifies the bulbocavernosus muscle, transverse perineal muscles, and perineal membrane with full understanding.	Explains the anatomical structures with minor errors or omissions.	Identifies the structures but lacks a clear understanding.	Fails to correctly identify or explain the anatomical structures.
7. Reapproximation of Vaginal Wall	Demonstrates excellent technique in reapproximating the vaginal wall and perineal trunk to their original positions.	Reapproximates the vaginal wall and perineal trunk with minor errors.	Reapproximation is attempted but lacks precision or proper alignment.	Fails to properly reapproximate the vaginal wall and perineal trunk.
8. Suture Selection	Selects the appropriate 3-0 or 2-0 absorbable suture confidently and correctly.	Selects the correct suture but lacks confidence in the choice.	Selects the correct suture but with hesitation or uncertainty.	Fails to select the correct suture, affecting the procedure.
9. Suture Technique	Executes continuous or interrupted sutures with precision, ensuring the torn ends are perfectly reapproximated.	Performs sutures effectively with minor gaps or interruptions.	Sutures are completed but with noticeable gaps or improper technique.	Sutures are poorly executed, with significant gaps or incorrect techniques.
10. Suture Application	Applies the suture as described, ensuring secure and accurate closure of the wound.	Applies the suture with minor deviations from the described method.	Suture application is inconsistent or deviates significantly from the described method.	The suture application is incorrect, leading to an insecure or incomplete closure.
11. Technique Application	Demonstrates full mastery of the episiotomy technique, following all steps accurately and confidently.	Applies the technique competently with minor errors or hesitation.	Applies the technique with noticeable errors or a lack of confidence.	Fails to apply the technique correctly, with significant errors or omissions.
12. Use of Materials	Uses all materials appropriately and efficiently, with no waste or misuse.	Uses materials appropriately with minimal waste or misuse.	Uses materials but with noticeable waste or minor misuse.	Fails to use materials appropriately, leading to waste or improper application.

Source: [7,8,15].

**Table 4 healthcare-13-01472-t004:** Personal information (n = 61).

Age	n	%
21 22	43 18	70.4 29.5
Living place		
City	31	50.8
District	20	32.8
Village	10	16.4
Economic situation		
My income is more than my expenses	12	20
My income is equal to my expenses	25	42
My income is less than my expenses	24	38
Access to the Internet		
I frequently have problems	6	9.8
Sometimes I have problems	20	32.8
Good	29	47.5
Pretty good	6	9.8
I have sutured a living tissue before		
Yes	14	23.0
No	46	75.4
I am indecisive	1	1.6
I have seen episiotomy suturization in the delivery room before		
Yes	23	37.7
No	38	62.3
I have performed episiotomy repair in the delivery room before		
Yes	3	4.9
No	58	95.1
Interest in clinical practice simulations		
Good	20	32.8
Very Good	41	67.2
	Average	Standard Dev.
Monthly Internet payment for courses	96.80	34.79
Latest Clinical Practice academic score	95.56	6.23

**Table 5 healthcare-13-01472-t005:** Descriptive statistics between SCLS and OLRS.

	Mean	Std Dev.	Min.	Max.
OLRS	74.96	8.12	51	90
SCLS	47.10	10.8	12	60
Student satisfaction with learning	20.24	4.67	5	25
Confidence in learning	26.86	6.46	7	35

SCLS (Student Satisfaction and Self-Confidence in Learning Scale); OLRS (Online Learning Readiness Scale).

**Table 6 healthcare-13-01472-t006:** The relationships between SCLS and OLRS.

		Online Learning Readiness
**Student Satisfaction**	r	0.204
	*p*	0.159
	n	61
**Confidence in Learning**	r	0.326
	*p*	0.022 *
	n	61

r: Pearson correlation coefficient; *: *p* < 0.05 (statistically significant); SCLS (Student Satisfaction and Self-Confidence in Learning Scale); OLRS (Online Learning Readiness Scale).

**Table 7 healthcare-13-01472-t007:** Comparisons of students’ answers to pre- and post-training questions.

		Yes		No		*p*	Effect Size (%95 C.I.)
		N	%	N	%		
Can an episiotomy study with a simulation made in living animal tissue increase your dexterity?	Before	54	88.5	7	11.5	0.125	1.145 (0.844–1.553)
	After	59	96.7	2	3.3		
Should an episiotomy study with simulation on living animal tissue be performed before a real episiotomy study?	Before	41	67.2	20	32.8	0.003	1.090 (0.890–1.336)
	After	55	90.2	6	9.8		
Does an episiotomy study with simulation on living animal tissue help you to imagine a real episiotomy study?	Before	52	85.2	9	14.8	1.000	1.662 (0.921–2.996)
	After	53	86.9	8	13.1		
I believe that this training improved my hand skills, even though it was an online education.	Before	30	49.2	31	50.8	0.000	0.999 (0.911–1.096)
	After	59	96.7	2	3.3		
I find myself incapable of suturing.	Before	28	45.9	33	54.1	0.004	8.250 (2.048–33.235)
	After	16	26.2	45	73.8		
I trust myself to open an episiotomy.	Before	27	44.3	34	55.7	0.815	2.393 (1.346–4.254)
	After	29	47.5	32	52.5		
I trust myself to repair an episiotomy.	Before	28	45.9	33	54.1	0.210	2.464 (1.476–4.114)
	After	34	55.7	27	44.3		
Learning episiotomy with online education is difficult.	Before	33	54.1	28	45.9	0.332	3.111 (1.471–6.581)
	After	28	45.9	33	54.1		

**Table 8 healthcare-13-01472-t008:** Non-parametric correlations between students’ performance rated by observers and students’ self-reported satisfaction, confidence, and final mark from the Clinical Practice module.

			Observed Performance During Training	Online Readiness Scale	Student Learning Satisfaction	Student Self- Confidence	Mark in Clinical Practice After Training
Spearman’s rho	Observed performance during training	rho	1.000	−0.172	−0.116	−0.067	0.327 *
		*p*		0.322	0.508	0.703	0.011
		N		35	35	35	60
	Online readiness scale	rho		1.000	0.393 **	0.496 **	0.053
		*p*			0.008	0.001	0.764
		N			45	45	35
	Student learning satisfaction	rho			1.000	0.844 **	0.178
		*p*				0.000	0.306
		N				45	35
	Student self-confidence	rho				1.000	0.052
		*p*					0.768
		N					35
	Mark in Clinical Practice after training	rho					1.000
		*p*					
		N					

* Correlation is significant at the 0.05 level (2-tailed); ** Correlation is significant at the 0.01 level (2-tailed).

## Data Availability

The dataset is available on request from the authors.

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
