# Peer review of "Enhancing Episiotomy Skills Through Interactive Online Simulation"

_healthcare, 2025, doi:10.3390/healthcare13121472_

Round 1
Reviewer 1 Report
Comments and Suggestions for Authors
The pandemic and the subsequent period have highlighted the importance of flexibility and adaptability in education systems. Research shows that interactive online simulations can maintain the continuity of education during exceptional times. Simulations help students develop their professional skills, stay motivated, and prepare for real-life situations. During and after the COVID-19 pandemic, these findings provide valuable insights for educators and policymakers.
This study clearly highlighted that the COVID-19 pandemic significantly limited the clinical training of midwifery students. Online simulation proved to be an effective alternative for developing essential clinical skills. This research evaluated the impact of online simulation training on episiotomy repair for students whose clinical practice was limited. The implementation of the study was highly innovative. Of course, there is a risk inherent in the setup, which the study points out, that not all students were able to prepare independently for the exercises in the same way. This certainly caused some variation in the results.
The study used a mixed-method approach, and the results were collected from 61 midwifery students. The abundance of methods and surveys was a factor that increased the diversity of the data. However, in terms of the readability of the article, it made the article heavy and somewhat confusing in parts. Readability would be improved if the description of the method, especially section 2.2, could be condensed or if a table or figure of the process could be provided.
There were quite a few research questions, and I wonder if they could have been condensed or narrowed down for the overall clarity of the article.
The reference list has not been systematically compiled according to a single citation technique. This needs to be corrected.
Author Response
The authors would like to thank to the Reviewers for their valuable comments because the comments will help to increase the quality of the study. The corrections are made accordingly and the answers for each comment is provided below.
RESPONSE TO REVIEWER 1 COMMENTS
Comments 1: The study used a mixed-method approach, and the results were collected from 61 midwifery students. The abundance of methods and surveys was a factor that increased the diversity of the data. However, in terms of the readability of the article, it made the article heavy and somewhat confusing in parts. Readability would be improved if the description of the method, especially section 2.2, could be condensed or if a table or figure of the process could be provided.
Response 1: . The manuscript in terms of grammar and necessary revisions are performed. In addition, Table 2 is added in Section 2 to improve the description of the method. Comments 2: There were quite a few research questions, and I wonder if they could have been condensed or narrowed down for the overall clarity of the article. Response 2: Table 2 is added to the manuscript in order to clarify the research questions.
Comments 3: References list and format must be organize.
Response 3: The reference list is revised accordinglyThe authors would like to thank to the Reviewers for their valuable comments because the comments will help to increase the quality of the study. The corrections are made accordingly and the answers for each comment is provided below.

Reviewer 2 Report
Comments and Suggestions for Authors
The article entitled: “Enhancing episiotomy skills through interactive online simulation” is a mix-method approach from 61 midwifery students.
There are some suggestions that can be taken into account:
1.Introduction: lines 55—56 are repeated in lines 57-58
- Material and Methods:
- please include a photo of the trainer
- line 116: please explain why there are 19 students not included in the analisis??
- please include an explanation of how teachers can improve the performance of the students
- Results:
- Table 3: please explain the meaning of SCLS and OLRS
- Discussion:
-line 377: do not start a sentence with a number (or write it with letters)
- Bibliography
- Please review references: 2,10,20,22,29
Author Response
RESPONSE TO REVIEWER 2 COMMENTS
Comments 1: 1.Introduction: lines 55—56 are repeated in lines 57-58
Response 1: The manuscript is revised accordingly,
Comments 2: Material and Methods: - please include a photo of the trainer
Response 2: Figure 1 is added as well as including the webpage of a short videos link.
Comments 3: line 116: please explain why there are 19 students not included in the analisis??
Response 3. The issue is discussed in section “2.1 Participants” by revising and adding the following text:
. Final-year students enrolled in a university midwifery program who voluntarily agreed to participate, had access to the standard equipment and conditions required for the practice sessions, and completed all phases of the study without interruption were included in the research. Participation was entirely voluntary, and students were informed of their right to withdraw from the study at any time without penalty.
Final-year students enrolled in a university midwifery program who voluntarily agreed to participate, had access to the standard equipment and conditions required for the practice sessions, and completed all phases of the study without interruption were included in the research. Participation was entirely voluntary, and students were informed of their right to withdraw from the study at any time without penalty.
Although a total of 80 students initially participated in the study, the analysis was conducted with data from 61 students who were able to perform the procedure on live tissue. The rationale for focusing on live tissue was to provide a more realistic and hands-on experience in episiotomy repair.
Nineteen students were excluded from the analysis as they did not meet the material-related eligibility criteria. These students, facing financial constraints, were unable to obtain live tissue materials and instead conducted the procedures using alternative resources such as sponges or synthetic leather. Nevertheless, their participation in the practical sessions was encouraged in accordance with the principle of equity in education.
It is noteworthy that approximately 80% of the participants belonged to middle or lower socioeconomic backgrounds, which may limit their access to appropriate training materials. This financial disparity highlights a significant challenge in ensuring uniformity in educational opportunities and may necessitate the exclusion of some students from data analysis to maintain methodological rigor. In line with this rationale, students who were unable to access the required materials were excluded from the final analysis.
The university’s midwifery department is only able to provide laboratory materials for in-person practical sessions and is not in a position to offer additional financial support or distribute materials to students individually.
Comments 4: please include an explanation of how teachers can improve the performance of the students
Response 4: The following text is added in section “5. Conclusions” to address the issue.
To improve student performance, educators should integrate interactive online simu-lations into the curriculum, especially when clinical training is limited. For instance, providing guided practice, timely feedback, and personalized support for students with low self-confidence can enhance both practical skills and clinical success.
Comments 5: Table 5: please explain the meaning of SCLS and OLRS
Response 5: The explanations are added to section “3. Results”.
SCLS(Student Satisfaction and Self-Confidence in Learning Scale), OLRS (Online Learning Readiness Scale).
Comments 6: -line 377: do not start a sentence with a number (or write it with letters)
Response 6: The sentence is revised in section “4. Discussion” accordingly.
A total of 96.7% of the students found the interactive online simulation training method effective and beneficial (Table 7).
Comments 7: Bibliography
Response 7: The bibliography of the authors is provided as follows:
Prof. Dr. Hava Özkan is a respected academic in the field of midwifery at Atatürk University, Turkey. She has significantly contributed to midwifery education, maternal health, and simulation-based learning. Her research focuses on maternal and newborn health, postpartum mental health, breastfeeding self-efficacy, and innovative teaching methods. She has published extensively in national and international journals and has led several interdisciplinary projects funded by TUBITAK and other institutions.
As head of the midwifery department, she played a key role in developing modern curricula and expanding simulation-based training. During the COVID-19 pandemic, she led the transition to digital midwifery education and emphasized the importance of clinical skill development through online tools. She is also active in national professional organizations and international collaborations. Known for her mentorship, she has guided many students into academic and clinical leadership roles. Her work continues to influence policy and practice in women's health and midwifery education.
Comments 8: Please review references: 2,10,20,22,29
Response 8: The references are revised accordingly.

Reviewer 3 Report
Comments and Suggestions for Authors
Dear Authors
Thanks for your study regarding online training
1-What were your exact inclusion and exclusion criteria?
2-Which platform did you perform your training and assessment on?
3-How were students graded?
4-How long did the simulation training last?
5-Who assessed students' performance?
6-It is better to summarize the conclusion part. It is too long
Author Response
RESPONSE TO REVIEWER 3 COMMENTS
Comments 1: What were your exact inclusion and exclusion criteria?
Response 1 Thank you for pointing this out. We agree with this comment. We revised again. Page 3. “Students in the final year of a university midwifery program who volunteered to participate, were able to access the standard equipment and conditions required for the practice process, and completed all phases of the study without interruption were included in the research”
Comments 2: How were students graded?
Response 2: The following text is revised/added in section “2.4”
Episiotomy Simulation Skills Rubric
The performance of the students is evaluated using a rubric where each criterion is graded on and pointed from 1 and 4, where the lowest score is 1 and the highest is 48. (Table 1).” Page 7.
Comments 3: Which platform did you perform your training and assessment on?
Response 3: The following text is revised/added in section “2.2.1 Creating an interactive online education environment
The trainer installed a camera at home and demonstrated how the visual feed would be streamed to students in real-time via the internet. These sessions were recorded through the Distance Education Module of the University Information Management System, which serves as an existing course delivery platform. This system was widely utilized by the university, particularly during the COVID-19 pandemic.
Comments 4: How long did the simulation training last?
Response 4: The following text is revised/added in section “2.2 Procedure”
“The 4 groups had interactive practice in separate periods, while the duration of each training was 60 minutes, total of training 240 minutes”.
Comments 5: Who assessed students' performance?
Response 5: The following text is revised/added in section “Table 2. Simulation training procedure – Flow and timeline.”.
3. Observation and Rubric Filling |
Each student’s performance was observed in real-time. |
During the session |
Two external experts and Two Academic Instructors |
Comments 6: It is better to summarize the conclusion part. It is too long.
Response 6: The Conclusion part is revised accordingly but the authors think that the current form aligns well with the main text's approach and properties.

Reviewer 4 Report
Comments and Suggestions for Authors
Line 96-107: it should be better if you put a table with this information, i would be more clear
Line 115: I don’t understand this sentence. What do you mean by students who volunteered were included? All from the 4th course or they could be from other coruses?, rewrite this sentence for more clear understanding.
Line 116-120: The university did not pay the material of the simulation?
Table 2: Why you ask about economic situation? Does it affects the results?
Author Response
RESPONSE TO REVIEWER 4 COMMENTS
Comments 1: Line 96-107: it should be better if you put a table with this information, i would be more clear
- Response 1: Thank you for your offer, we added a tablet o page 3. The name of the table 1. The questions were directed to the online training participants
Comments 2: Line 115: I don’t understand this sentence. What do you mean by students who volunteered were included? All from the 4th course or they could be from other coruses?, rewrite this sentence for more clear understanding.
Response 2: Thank you for your offer. We explained previously this section. We didn’t revise again because Even if they were senior students, they might not want to participate in this practice, and we couldn't force them”.
The issue is discussed in section “2.1 Participants” by revising and adding the following text:
Final-year students enrolled in a university midwifery program who voluntarily agreed to participate, had access to the standard equipment and conditions required for the practice sessions, and completed all phases of the study without interruption were included in the research.
Comments 3: Line 116-120: The university did not pay the material of the simulation?
Response 3: The university partially pays for the expenses. The issue is discussed in section “2.1 Participants”
The university’s midwifery department is only able to provide laboratory materials for in-person practical sessions and is not in a position to offer additional financial support or distribute materials to students individually.
Comments 4: Table 2: Why you ask about economic situation? Does it affects the results?
Response 4: The demographic characteristics of the students may affect the results obtained in the study. They may not have access to the internet and practice materials. In this case, it may be necessary to exclude them from the study or the analysis, which has shown that economic status is an important factor in the results reached in this study. Therefore, I added a sentence to the study to explain the point you made. The issue is discussed in section “2.1 Participants”
It is noteworthy that approximately 80% of the participants belonged to middle or lower socioeconomic backgrounds, which may limit their access to appropriate training materials. This financial disparity highlights a significant challenge in ensuring uniformity in educational opportunities and may necessitate the exclusion of some students from data analysis to maintain methodological rigor. In line with this rationale, students who were unable to access the required materials were excluded from the final analysis.
